# Observation of nanoscale magnetic fields using twisted electron beams

Vincenzo Grillo[1,2], Tyler R. Harvey [3], Federico Venturi [1,4], Jordan S. Pierce[3], Roberto Balboni [5], Frédéric Bouchard[6], Gian Carlo Gazzadi[1], Stefano Frabboni [1,4], Amir H. Tavabi[7], Zi-An Li[7,8], Rafal E. Dunin-Borkowski[7], Robert W. Boyd[6,9,10], Benjamin J. McMorran [3] & Ebrahim Karimi [6,11]

Electron waves give an unprecedented enhancement to the field of microscopy by providing higher resolving power compared to their optical counterpart. Further information about a specimen, such as electric and magnetic features, can be revealed in electron microscopy because electrons possess both a magnetic moment and charge. In-plane magnetic structures in materials can be studied experimentally using the effect of the Lorentz force. On the other hand, full mapping of the magnetic field has hitherto remained challenging. Here we measure a nanoscale out-of-plane magnetic field by interfering a highly twisted electron vortex beam with a reference wave. We implement a recently developed holographic technique to manipulate the electron wavefunction, which gives free electrons an additional unbounded quantized magnetic moment along their propagation direction. Our finding demonstrates that full reconstruction of all three components of nanoscale magnetic fields is possible without tilting the specimen.

[1] CNR-Istituto Nanoscienze, Centro S3, Via G Campi 213/a, I-41125 Modena, Italy. [2] CNR-IMEM Parco Area delle Scienze 37/A, I-43124 Parma, Italy. [3] Department of Physics, University of Oregon, Eugene, OR 97403-1274, USA. [4] Dipartimento FIM, Universitá di Modena e Reggio Emilia, Via G. Campi 213/a, I-41125 Modena, Italy. [5] CNR-IMM Bologna, Via P. Gobetti 101, 40129 Bologna, Italy. [6] Department of Physics, University of Ottawa, 25 Templeton St., Ottawa, ON, Canada K1N 6N5. [7] Ernst Ruska-Centre for Microscopy and Spectroscopy with Electrons and Peter Grünberg Institute, Forschungszentrum, Jülich 52425, Germany. [8] Institute of Physics, Chinese Academy of Sciences (CAS), Beijing 100190, China. [9] Institute of Optics, University of Rochester, Rochester, NY 14627, USA. [10] School of Physics and Astronomy, University of Glasgow, Glasgow G12 8QQ, UK. [11] Department of Physics, Institute for Advanced Studies in Basic Sciences, Zanjan 45137-66731, Iran. Correspondence and requests for materials should be addressed to E.K. (email: ekarimi@uottawa.ca)

Vortices can be observed in different forms spanning nano to astronomical scales, such as dust devils, whirlpools, and rotating black holes[1, 2]. A vortex can be characterized by its circulation quantity around a singular point, which is located at its center. This circulation around such a point may lead to an intrinsic quantity, referred to as orbital angular momentum (OAM)[3]. In the quantum regime, any particle owning a spiraling phase structure $\exp(i\ell\varphi)$ carries a well-defined quantized OAM value of $\ell\hbar$ along the $z$-axis[4], where $\ell$ and $\hbar$ are an integer and the reduced Planck constant, respectively, and $\rho$, $\varphi$, and $z$ are cylindrical coordinates. For free particles possessing a net charge, for example electrons, such circulations result in an orbiting current density probability, which produces a quantized magnetic moment $\ell\mu_B$ along the particle propagation direction[5], where $\mu_B$ is the Bohr magneton. Apart from abstract fundamental interests, optical vortices have found potential applications in applied sciences, such as data transmission[6], quantum information[7], and secure computation[8, 9].

The resolution of any imaging system is fundamentally limited and is determined by the wavelength of the illumination $\lambda$ and the numerical aperture (NA) of the imaging lens, that is, $\lambda/(2\text{NA})$, where $\text{NA} = \sin\theta$ and $\theta$ is the acceptance angle of the lens. This is known as Abbe's law. Therefore, images with higher spatial resolutions need to be obtained by shorter wavelengths, including X-ray and particle, such as electron beams[10, 11]. The de Broglie wavelength $\lambda_{dB}$ associated with commercially available electron beams is five orders of magnitude smaller than that of visible light, which allows resolving very fine structures. Further structural information, such as electric and magnetic features[12, 13], can be measured since an electron possesses a negative charge $e = -|e|$ and a magnetic moment of $\pm\mu_B$[5].

For instance, as illustrated in Fig. 1a, b, an in-plane magnetic field (that is, magnetic flux density) induces a transverse Lorentz force, which results in a beam deviation. By knowing the energy of the electron beam and the beam deviation, the in-plane magnetic field can be obtained experimentally. The out-of-plane magnetic field, however, cannot be measured by this method without tilting the specimen because the induced Lorentz force due to a magnetic field parallel to the electron beam propagation direction is zero[14, 15], see Fig. 1c. Thus, the electron beam traverses without being significantly affected by the parallel magnetic field, as spin and lensing effects are both negligible. Twisted electrons possessing an unbounded magnetic moment $\ell\mu_B$ along their propagation direction can therefore undergo interactions with out-of-plane magnetic fields, and thus have the potential to explore magnetic structure[16]. Both the spatial and the temporal coherence of electron beams, however, must be sufficiently controlled for performing such experiments. Recent experimental advances make electrons with exotic spatial probability and phase structures[17, 18] such as electron vortex[19–22], Bessel[23], and Airy[24] beams, feasible.

Here we characterize the nanoscale magnetic field of a magnetic pillar using an electron vortex beam carrying an OAM value of $\pm 200\hbar$. In particular, the out-of-plane magnetic field induced by the nano-pillar introduces a global phase shift onto the electron vortex beam, which is measured using an interferometric scheme. From the measured phase shift, the longitudinal magnetic field is then be determined. Our results allow for the full reconstruction, transverse and longitudinal, of the magnetic properties of nanoscale specimens.

## Results

**Interaction of an electron vortex beam with a magnetic dipole**. The interaction of non-relativistic structured electrons $\psi(\boldsymbol{r})$ with a magnetic field $\mathbf{B} = \nabla \times \mathbf{A}$, in general, can be described by the Pauli–Schrödinger equation, where $\mathbf{A}$ is the vector potential. We consider a case where the electron interacts with a localized magnetic dipole that is oriented along the electron beam propagation axis $z$, that is, $\mathbf{A} = A(\mathbf{r})\,\mathbf{e}_\varphi$, where $\mathbf{e}_\varphi$ is the azimuthal unit vector. A scanning electron microscope image of a nanofabricated magnetized nanowire (magnetic dipole), hereafter referred to as a magnetic pillar, is shown in Fig. 2a. The pillar dimensions were ~220 nm in diameter, 970 nm in height, and it was fabricated by focused electron-beam-induced deposition (FEBID) of a co-carbonyl $Co_2(CO)_8$ precursor, producing a nanostructure with a very strong localized magnetic field[25] (see "Methods" section for fabrication details). The generated magnetic field of the pillar forms a physical dipole, where its value inside the pillar reaches typically $\simeq 1$ Tesla. A magnetic induction map of the pillar recorded using off-axis electron holography is shown in Fig. 2b. The case studied above can be extended to reveal the magnetic properties of an unknown specimen. This relies on the fact that magnetic features can be considered as an array of structural localized magnetic dipoles possessing different strengths.

Although magnetic monopoles have been simulated in different experiments[26, 27], the search for natural magnetic



**Fig. 1** Schematic illustration of the effect of the Lorentz force on an electron beam. The induced Lorentz forces due to in-plane uniform magnetic fields $B_x$ and $B_y$, **a** and **b**, that are given by $F_y = -|e|v_z B_x$ and $F_x = -|e|v_z B_y$ deviate the incoming electron beam in the $x$ and $y$ directions, respectively. When the magnetic field is parallel to the beam propagation direction, **c**, the Lorentz force vanishes, and thus the electron beam propagates without being affected by the out-of-plane magnetic field. Small effects such as lensing and spin interactions on the electron beam wavefunction are neglected

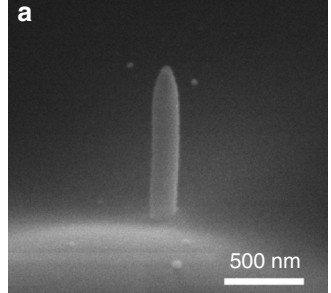
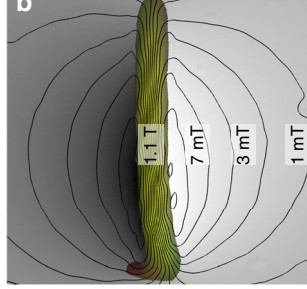

**Fig. 2** Scanning electron microscope image and magnetic field distribution of the pillar. **a** The magnetic pillar dimension is about 220 nm in diameter and 970 nm in height. The pillar was grown by electron-beam-induced deposition. The magnetic field at the center of the pillar reaches values as high as 1.1 T as shown in **b**. **b** Magnetic field distribution of a fabricated pillar measured using electron holography when it is tilted by about 60°—note that an in-plane magnetic field induces a transverse Lorentz force (see Fig. 1a, b). The *lines* are equiphase contours (here drawn every $\pi$-phase steps), and correspond to field lines of the magnetic field

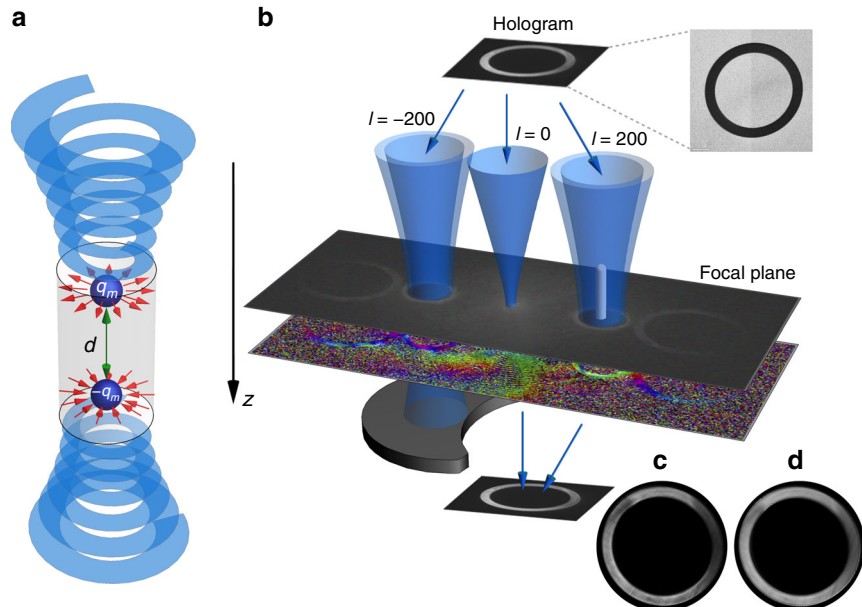

**Fig. 3** Schematic illustration of the set-up used for measuring an out-of-plane magnetic field. **a** Schematic diagram of the interaction of a twisted electron beam with a magnetic dipole comprised of two magnetic monopoles separated by a distance $d$ with opposite charges of $q_m$ and $-q_m$. The *red arrows* represent the magnetic fields induced by the magnetic monopoles. **b** An amplitude-phase hologram that is comprised of an annular ring with a structured pitch-fork mask—its energy filtered transmission electron microscope image is shown in the *inset*—is illuminated by a beam of electrons. .Scale bar represents 5 μm. Only electrons traversing throughout the annular region gain phase, and are absorbed by the mask elsewhere. The 0th, 1st, and $-1$st order diffracted electrons carry OAM values 0, 200, and $-200$, respectively. The diffraction orders become spatially distinct in the focal plane, where experimentally measured probability and phase distributions for the electron are shown in the *top* and *bottom plane*, respectively. Therefore, they constitute two arms of an interferometric-type apparatus. A magnetic pillar generating a longitudinal magnetic field along the propagation direction of the electrons acts as a phase shifter, given that the first-order diffracted electron carries a high OAM. The beams are then coherently recombined in the image plane of the hologram where the interference patterns are shown **c** with and **d** without the pillar. The interference of the 0th, 1st ($-1$st) orders is recorded, and a shift in the fringes of the interference pattern is observed as the magnetic pillar is brought in and out the path of the 1st ($-1$st) order diffracted electron

monopoles has so far been unsuccessful[28]. However, it is mathematically convenient to implement it in our interaction analysis. Therefore, we consider that the magnetic dipole comprises two monopoles having opposite magnetic charges, $q_m$ and $-q_m$, that are kept at a distance $d$ from each other, see Fig. 3a. Upon interacting with a magnetic monopole, an electron acquires a coordinate-dependent U(1)-phase, $\alpha = \frac{e}{\hbar} \int \mathbf{A} \cdot d\mathbf{x}$, which is referred to as an Aharonov-Bohm phase[29], where $d\mathbf{x}$ is the infinitesimal part of the electron trajectory. For a specific case, this phase can be a multiple of $2\pi$ in the transverse plane around the monopole, thus $2\pi\nu = e\Phi/\hbar$, where $\nu$ is the topological charge, $\Phi = \int \mathbf{B} \cdot d\mathbf{s} = 2\pi a^2 B$ is the magnetic flux and $a$ is the radius of a transverse loop surrounding the object of interest. Although generally $\nu$ is a real number, here it is assumed to be an integer and is referred to as the topological charge of the monopole.

Recently, it was shown that vortex beams may be generated based on interaction with magnetic monopoles[30]. Thus, magnetic monopoles act as OAM ladder operators: depending on the geometry and monopole charges, they either increase or decrease the initial electron OAM state by $\nu = e\Phi/(2\pi\hbar)$, that is, $\ell \xrightarrow{\text{monopole } q_m} \ell + \nu$ (Supplementary Note 1). Theoretical calculations show that such an interaction, because of the cylindrically symmetric geometry, preserves canonical OAM[31–33]. Nonetheless, it changes the kinetic OAM of the electron beam, which produces a similar effect in the far-field region (Supplementary Note 2).

A peculiarity of electrons carrying OAM is their propagation-dependent phase shift known as the Gouy phase shift[32, 34], that is $\Theta_{\text{Gouy}} = (|\ell| + 1)\arctan(z/z_R)$, where $\ell$ is the OAM value of the electron, $z$ is the propagation distance, $z_R = \pi w_0^2/\lambda_{\text{dB}}$ is the Rayleigh range, and $w_0$ is a beam parameter given by the beam

radius at the waist[34]. In the presence of a magnetic monopole, the increase in OAM thus results in an increase of the Gouy phase. In the case of a small propagation distance $d$ and a beam radius $\rho_m$, corresponding to the radius of maximum intensity, the Gouy phase after passing through the monopole is approximately given by $\Theta_{\text{Gouy}}(\ell) \approx \frac{d\lambda_{\text{dB}}}{\pi\rho_m^2}(\ell + \nu)^2$. Hence, an electron with a positive OAM value passing through a magnetic monopole gains a relative phase of $\Theta^+ = \Theta_{\text{Gouy}}(\ell)$ with respect to a reference beam. Similarly, an electron with a negative OAM value gains a relative phase of $\Theta^- = \Theta_{\text{Gouy}}(-\ell)$. Finally, the relative phase between the positive and negative OAM components will be given by $\Delta\Theta = \Theta^+ - \Theta^- \approx \left(\frac{d\lambda_{\text{dB}}}{\pi\rho_m^2}\right)4\ell\nu$, where $\rho_m$ may be adjusted independently of $\ell$. Once the electron passes through the second monopole, the OAM value is decreased by a factor of $\nu$, thus returning to its original OAM value, that is, $\ell + \nu \xrightarrow{\text{monopole} - q_m} \ell$. Hereafter, no more propagation-dependent phase is acquired by the electrons. In addition, the transverse fringing field introduces a non-negligible lensing effect, which is independent of the beam OAM value.

**Experimental characterization of a magnetic pillar with an electron vortex beam.** The $\varepsilon = 300$ keV energy electrons generated by a field-emission source in a transmission electron microscope (TEM) possess a relatively high transverse coherence length, making them suitable for our experiment. The electron beam, with an associated de Broglie wavelength of $\lambda_{\text{dB}} = 1.97$ pm, is sent to an amplitude-phase hologram. The amplitude-phase hologram, which is a transparent annular aperture with a phase structure (forming a pitchfork shape with 200 phase dislocations; a TEM image of the hologram is shown in Supplementary Fig. 1), is used to convert the truncated plane wave of a TEM into

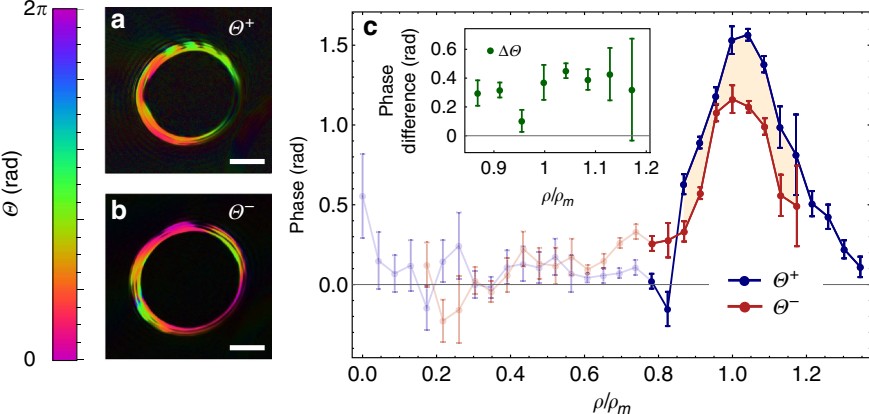

**Fig. 4** Experimentally observed phase shift of twisted electron beams upon interaction with a magnetic pillar. **a** and **b** show the digitally reconstructed phase shift for beams carrying OAM values of + 200 and −200, respectively. Scale bar represents 100 nm. **c** The *points* indicate the experimental data for measured relative phase changes that are obtained by the electron interferometer shown in Fig. 3. The *error bars* are based on standard deviations in different sets of measurements. *Red* and *blue colors* are the measured phases for beams carrying positive $\Theta^+$ and negative $\Theta^-$ OAM values of 200, that is, $\ell = \pm 200$. The relative phase $\Delta\Theta$ is illustrated by the *beige shaded area*. These data were evaluated for regions close to $\rho_m = 0.16$ m radius. Semi-transparent zones indicate regions where the probability of finding electrons is zero. The *solid lines* are obtained by interpolation

a twisted electron beam (see "Methods" section for more details). After traversing through the hologram, the wavefunction of the electron gains a phase and probability distribution dictated by the hologram structure. Since the hologram possesses a lateral grating, electrons diffract into discrete orders at angles set by the grating pitch and the electron de Broglie wavelength. However, each order of diffraction has a different transverse phase structure. For the fabricated hologram, which is shown in Fig. 3b in the inset, the 0, 1st, and −1st orders of diffraction gain OAM values of 0, $200\hbar$, and $-200\hbar$, respectively. In addition to the phase, since the electron distribution just after the hologram is an annular ring, its Fourier transform, the far-field spatial distribution of the electron, results in a Bessel-like function. Thus, the beam in the first orders of diffraction, in the far-field of the hologram, is approximately a Bessel beam carrying an OAM high-quantum value of $\pm 200\hbar$[23].

The grating pitch period is engineered so as to avoid any spatial overlap between the zeroth and first order in the diffraction plane of the hologram, which coincides with the specimen plane. In order to pass two dominant active beams, an aperture is placed in the specimen plane. This aperture blocks unneeded diffraction orders such that the zeroth- and the first order, namely the reference and signal beams, are the strongest remaining beams (Supplementary Fig. 2). A magnetic lens is then used to recombine and image the zeroth- and the first order of diffraction in the image plane of the hologram, which is also the diffraction plane of the specimen. The set-up, which is shown in Fig. 3b, is analogous to a two-arm interferometer for a single electron, where the electron wavefunction of the reference and signal arms have 0 and $\pm 200\hbar$ OAM, respectively. The interference pattern in the image plane of the hologram—the diffraction plane of the specimen—is a ring of black and white fringes with 200 dislocations. This is due to the fact that the electron wavefunction at the reference arm carries an OAM of 200 with respect to the signal beam. Hence, such a set-up is capable of revealing any induced relative phase changes $\Theta^+$ between the two beams. The relative phase change between the two arms results in a shift of the fringe pattern in the image plane of the hologram, where a $\pi$-phase change shifts the bright fringes to dark and vice versa. The relative phase shift between the beam carrying OAM and the reference beam is obtained by performing a Fourier transform of the interference pattern images in different cases. See

Supplementary Note 4 for more technical details on the calculation of phase shift from recorded holograms.

We placed the magnetic pillar at the center of the signal beam, that is the electron Bessel beam that possesses OAM. This was achieved in steps of less than 10 nm, in order to move the pillar in the specimen plane of the TEM. The probability of finding the electron at the beam center is zero, since its wavefunction is singular at the beam origin. The signal beam interacts with the pillar without being absorbed by the carbon substrate, while its wavefunction gains a phase $\Theta^+$ upon elastic interaction with the magnetic field of the pillar. The phase $\Theta^+$ is global, and cannot be observed for a solo electron beam. However, interfering the signal with a reference beam reveals this global phase. The interference pattern in the image plane of the hologram—the diffraction plane of the specimen—was recorded for two different cases: with the magnetic pillar located at the center of a positive OAM beam (the first order of diffraction) and a negative OAM beam (the −1st order of diffraction). These two cases were then repeated on plain substrate with no magnetic material near the beam. The presence of the magnetic field shifts the position of the bright and dark fringes for both cases (Supplementary Fig. 3). Figure 4a, b shows the measured phase shift on the image screen for these two different cases, that is $\Theta^+$ and $\Theta^-$. An average phase shift of $\overline{\Delta\Theta} = 0.34 \pm 0.03$ radian is observed, as shown in Fig. 4c, for the magnetic pillar shown in Fig. 2a. This phase shift corresponds to an average magnetic field $\overline{B}_z = 0.97 \pm 0.14$ T that is generated by the pillar over the interaction length of 970 nm. In order to put our technique to the test, we fabricated a second pillar specimen. The dimensions of the second pillar were 170 nm in diameter and 1200 nm in height. We experimentally observed a phase shift of $\overline{\Delta\Theta} = 0.24 \pm 0.03$ radian for the second pillar, which corresponds to a magnetic field of $\overline{B}_z = 0.93 \pm 0.17$ T (Supplementary Figs. 4 and 5). The measured out-of-plane magnetic fields are in good agreement with the previously reported values of magnetic fields inside these nanowires[35], and the holographic measurement of the in-plane field when the pillar is tilted to 60°, as shown in Fig. 2b, that is $B = 1.1$ T. The value of the magnetic field for these nanowires is to some extent height dependent, but becomes saturated after a certain length, where the saturation magnetic field is around 1 T. The present sensitivity of the technique is about $10^8 \mu_B$, however, it can be increased by implementing electrons carrying larger OAM values.

## Discussion

Energy-loss magnetic circular dichroism (EMCD) has been introduced and exploited as a method to reveal magnetic chirality of materials, where the cross-section of inelastic interaction of electron beams with a material depends on its magnetization state[36]. EMCD can be evaluated by analyzing the OAM state of the inelastically scattered electrons[20]. However, dictated by the selection rule, only electron vortex beams possessing OAM values of $\ell = 0$ and $\ell = \pm 1$ can contribute to the dipole transition, and thus the technique is limited to low-OAM carrying beams within the dipole approximation. Furthermore, methods based on elastic scattering have also been proposed or demonstrated as possible probes of out-of-plane magnetic fields. In particular, the Zeeman interaction of the electron's OAM with the out-of-plane magnetic field in a magnetic material induces a change in the diffraction pattern of the scattered electron[37, 38]. Nonetheless, at the atomic resolution level, using low values of OAM, this technique suffers from detection limitations, but may be feasible with high OAM values. Although in several cases, image rotation has been associated with an out-of-plane magnetic field[32, 33, 39, 40], the experiment we present here is the first demonstration of quantitative measurement of an out-of-plane magnetic field with an electron vortex beam. The interferometric approach we present is a form of off-axis holography and has the advantage of the possibility to quantify and subtract off in-plane magnetic and electrostatic effects. Alternatively, image rotation-based methods offer an otherwise similar sensitivity to out-of-plane magnetic fields. The dynamical scattering approach[37, 38] offers a different contrast mechanism and, if experimentally demonstrated, may be a promising alternative. Unlike the technique we demonstrate, this method requires no reference and relies on the combined effect of diffraction and Larmor rotation to provide sensitivity to out-of-plane fields.

A further alternative approach would be to use beams that are a coherent superposition of two beams carrying opposite OAM values resulting in petal beams[32]. Upon interaction of these petal beams with a longitudinal magnetic field, each component gains a different phase, that is, $\Theta^+$ or $\Theta^-$. Therefore, the petal's orientation rotates due to the interaction with the longitudinal magnetic field[33, 39, 40]. This type of rotation might be interpreted as Larmor rotation, as previously observed for a constant magnetic field[39]. Nonetheless, the latter approach may be practically difficult, since it requires full isolation of the petal beam from the other diffraction orders. Moreover, the (transverse) kinetic momentum of the petal beams would be altered, which results in beam deviation when the beam interacts with a slowly space-varying longitudinal magnetic field. Both techniques can potentially be used to measure the magnetic structure of materials, specifically the out-of-plane component of the magnetic field. The petal beam approach can be used to map (by rastering) continuous thin films, while, in the interferometric scheme, electrons do not touch the material directly. Thus, our method may open up an avenue to explore magnetic features in dose-sensitive nanostructures, as well as superparamagnetic nanostructures whose fields are stabilised by the microscope objective lens.

## Methods

**Experimental set-up.** The primary experiment was performed at the University of Oregon on an FEI Titan microscope equipped with a Schottky field emitter, image aberration corrector, and GIF energy filter. The microscope was operated at 300 keV. For the experiments, a SiN hologram fabricated by FIB was introduced into the C2 condenser aperture plane of the microscope. We then turned the microscope to the so-called low mag mode, with the main objective lens turned to a low strength. Preliminary experiments have also been performed in CNR-IMM Bologna using an FEI Tecnai microscope operated at 200 keV with a similar configuration. The hologram placed in the specimen position produced a

diffraction pattern consisting of a central spot and different diffraction orders. The 1st and −1st diffraction orders possess OAM values of $\ell = 200$ and $\ell = -200$, respectively. We then used an objective aperture, which for this specific case was close to the plane of the specimen, to exclude from transmission all beams except the zeroth order, and one of the 1st and −1st orders. The interference pattern, in the image plane of the hologram, between these two beams was then digitally recorded and reconstructed by Fourier transform and reference phase subtraction to obtain a phase map (Supplementary Figs. 6 and 7). In the specimen plane, that is in the diffraction plane of the hologram, the signal beam was placed around a vertical magnetic pillar. As an additional reference, an image of the hologram on the substrate with no magnetic material near any beams was also recorded. A precise alignment of the images, to single-pixel precision, was used to remove the gradient on the (Fourier) phase difference in the vortex region. Residual transverse forces (e.g., due to in-plane magnetic fields) can introduce additional constant phase terms in this process. However, when performing an azimuthal average and subtraction, the effect of these forces vanishes (Supplementary Note 3).

**Nanowire fabrication and characterization.** The deposition of the magnetic pillars took place inside a Dual Beam (DB) system, the FEI Strata DB 235M, by means of FEBID. This technique, combining a gas injection system (GIS) and the electron beam of a scanning electron microscope, allows for the deposition of nanostructures of desired size and shape. The secondary electrons, which are generated by the interaction of the primary electron beam with the substrate, dissociate the precursor gas, leading to deposition of the desired element. In this experiment, the GIS was operated at room temperature and was mounted at a polar angle of 52° and an azimuthal angle of 115° with respect to the specimen surface. An injection nozzle with a reduced diameter of 160 m was installed in order to limit the pressure bursts that typically occur for this kind of precursor at the first opening, and to reach a gas pressure in the chamber on the order of $3 \times 10^{-6}$ mbar (with respect to a base pressure of $6 \times 10^{-7}$ mbar), a value that allowed for a fine control of the deposition process. The nozzle-to-specimen distance during deposition was about 200 m. A $Co_2(CO)_8$ precursor gas was used, resulting in the deposition of a Co with 75–85% purity in nanogranular form, corresponding to a saturation magnetization close to that of pure cobalt. The pillar's magnetic field was measured using off-axis electron holography in an FEI Titan TEM operated in Lorentz mode at Ernst Ruska-Centre Forschungszentrum Jülich. The electrostatic contribution to the measured phase was removed by acquiring a second hologram after inverting the specimen.

**Data availability.** All relevant data are available from the authors on reasonable request.

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

## Acknowledgements

The authors thank Jan Caron for fruitful discussions about magnetic field reconstruction. V.G. acknowledges the support of the Alexander von Humboldt Foundation. F.B. acknowledges the support of the Vanier Canada Graduate Scholarships Program of the Natural Sciences and Engineering Research Council of Canada (NSERC). S.F. and F.V. acknowledge the support of the University of Modena and Reggio Emilia (FAR 2015). R.E.-D. thanks the European Research Council for funding under the European Union's Seventh Framework Programme (FP7/2007-2013)/ERC grant agreement number 320832. We gratefully acknowledge the use of CAMCOR facilities, which have been purchased with a combination of federal and state funding. T.H. and B.M. are supported by the U.S. Department of Energy, Office of Science, Basic Energy Sciences, under Award DE-SC0010466. E.K. acknowledges the support of the Canada Research Chairs (CRC) Program. F.B., R.W.B., and E.K. acknowledge the support of the Max Planck–University of Ottawa Centre for Extreme and Quantum Photonics.

## Author contributions

V.G., R.W.B., B.J.M., T.R.H., and E.K. conceived the idea; V.G., T.R.H., R.B., and S.F. designed the experiment and performed the experiment; V.G. analyzed the data; J.S.P. and G.C.G. fabricated the hologram; F.V., G.C.G., and S.F. fabricated the magnetic pillar; A.H.T., Z.-A.L., and R.E.D.-B. tested the magnetic pillar. V.G., F.B., and E.K. developed the theory, and wrote the manuscript with the help from other authors. All authors discussed the results and contributed to the text of the manuscript.

## Additional information

**Competing interests:** The authors declare no competing financial interests.

