## [Peer Review File · Nature Communications]

Reviewers' comments:

Reviewer #1 (Remarks to the Author):

This paper reports the measurement of the out-of-plane magnetic fields of the magnetic nanopillar using the twisted electron vortex beams. The proposed method uses an interference between a twisted electron vortex beam and a non-twisted electron beam. On the other hand, conventional electron holography measures in-plane magnetic fields using an interference between non-twisted electron beams. Overall, I have a reliable impression. However, some questionable and insufficient points arise. Some problems must be removed prior to recommend for its publication in Nature Communications.

(1) The authors mentioned in page 3, "Twisted electrons, since they possess an unbounded magnetic moment μ_B along their propagation direction, can be put into interactions with out-of-plane magnetic fields,". Does the presented method use directly magnetic moments of twisted electron vortex beams to extract out-of-plane magnetic fields or mere phase shift? Do Landau and Larmor trajectory rotations of electrons about the magnetic field direction occur in the present case? Some experimental measurements involving the interactions between the twisted electron vortex beams and magnetic fields have already been reported (ref 7: Nature, 467, 1839, (2010), ref 26: Nature Commun., 5, 4586 (2014) and ref 31: Nature Physics, 10, 26 (2014).) The differences and advantages in more detail should be included in the manuscript.

(2) Relating to the above 1), in page 4, the authors presented the Gouy phase after passing through the monopole, Θ_{Gouy} . The authors should add the theoretical framework and the detailed derivation of Θ_{Gouy} in Methods. Otherwise we cannot evaluate whether the small propagation distance d is a correct assumption and the effectiveness of this method.

(3) The necessity of the high OAM (200) is unclear. The larger the beam OAM value is, the larger the beam size is. Θ_{Gouy} is inversely proportional to the square of a beam radius ρ_m . Also in page 5, the authors mentioned, "The signal beam interacts with the pillar without being absorbed by the carbon substrate,". Is the signal beam size larger than the magnetic pillar dimension, 220 nm in diameter? What is ρ_0 in the Fig.4 caption? We are interested what is the smallest possible signal beam size in this method? In addition to responding to these questions in the manuscript, the authors should add the actual beam radius ρ_m and the singular vortex core size.

(4) In page 6, the authors mentioned, "The presence of the magnetic field shifts the position of the bright and dark fringes for both cases." This is a very key data to give experimental confidence and see the trajectory rotations. The experimental fringe patterns to be distinctly visible should be included in Methods, in addition to the the transverse intensity distributions of the signal beam before-after passing through the magnetic pillar prior to interfering.

(5) An electromagnetic lens employs a strong longitudinal magnetic field. The authors should mention the influence on the measurements and the samples, and the experimental set-up (including Lorentz mode or not, the longitudinal magnetic field of the used lens) in the manuscript.

(6) The magnetic field distribution by the pillar is not uniform. The influence should be mentioned in the manuscript.

Reviewer #2 (Remarks to the Author):

Authors present a new method of measurement of out-of-plane magnetic fields using electron vortex beams in transmission electron microscopy. The experimental results are new and in my opinion convincing. Method can have a strong impact in the field of nano-magnetism and as such I believe that the manuscript is suitable for Nature Communications. I have however some concerns that authors need to address, before the manuscript can be published. They are listed in order of occurrence in the manuscript.

* introductory paragraph: (language) "having I intertwined" -> perhaps find a clearer, more explicit expression?

* page 2, last line: " $\lambda/(2NA)$ " -> the "NA" was defined only as an acronym. How should it be interpreted in a mathematical expression?

* page 3, paragraph discussing interaction of a an electron wave with magnetic field omits important references to

Edstrom et al., PRL 116, 127203 (2016)

Edstrom et al., PRB 94, 174414 (2016)

which deal with elastic interaction of electron vortices with magnetic fields in samples - a primary topic of this manuscript. These works must be put into context of the present work, to highlight parallels and differences of the method used by authors and the one proposed in the above-mentioned works.

* page 3, last two lines - some references highlighting the debate of the existence of magnetic monopole would be suitable here

* page 4: please define the exact meaning of the "transverse loops" in the definition of "a"

* page 4: a reference for the Guoy phase would be suitable at the place, where it is introduced

* page 4: authors should define the "Rayleigh range" to simplify reading for a reader not familiar with this concept. Alternatively, a suitable reference would be helpful.

* page 4 (language): "See Method for more details" -> "See Methods for more details"

* page 5: "is a Bessel beam..." - I think it would be fair to write rather "is approximately a Bessel beam...", because there is a non-negligible finite range of radii passing through the aperture

* page 5: the 2nd paragraph ("The grating pitch...") I found far too dense. Particularly its second half starting with "The interference pattern...". I would encourage the authors to expand this paragraph and provide a more detailed explanation that would be more accessible to the reader.

* page 6 (language): "almost height dependent" -> perhaps rather "to some extent height dependent" or something similar?

* page 6: "sensitivity of the technique is about $10^8 \mu_B$ " -> authors should explain, how was this estimated

* page 11: can authors elaborate, under what assumptions/conditions does the effect of residual transverse forces vanish in the azimuthal average? Are these conditions fulfilled for the presented experimental setup?

In addition, I would encourage authors to show the results of measurements for the other pillar quoted in the text - perhaps in Supplementary Information or as a new panel in Fig.4. Also, a more explicit description of the quantification of the average magnetic field in the sample would be helpful for researchers, which might want to apply this new technique in their experiments.

We would like to thank all reviewers for carefully reading our manuscript as well as the valuable comments and feedback which has helped us to improve the quality of our current manuscript.

Reviewer 1 - This paper reports the measurement of the out-of-plane magnetic fields of the magnetic nanopillar using the twisted electron vortex beams. The proposed method uses an interference between a twisted electron vortex beam and a non-twisted electron beam. On the other hand, conventional electron holography measures in-plane magnetic fields using an interference between non-twisted electron beams. Overall, I have a reliable impression. However, some questionable and insufficient points arise. Some problems must be removed prior to recommend for its publication in Nature Communications.

Authors' Response: we thank Reviewer 1 for carefully reading our manuscript and for their insightful suggestions. Below, please find our reply and modifications to the main text addressing Reviewer 1's inquiries. These comments helped us improve the quality of our manuscript.

(1) The authors mentioned in page 3, "Twisted electrons, since they possess an unbounded magnetic moment μ_B along their propagation direction, can be put into interactions with out-of-plane magnetic fields,". Does the presented method use directly magnetic moments of twisted electron vortex beams to extract out-of-plane magnetic fields or mere phase shift ?

Reply) The interference method we describe measures a phase shift. We have decided to adopt a model considering a phase shift due to the Gouy phase acquired by a twisted electron beam. From this model and the independently measured geometry of the pillar, we can reconstruct the flux through the pillar, or, equivalently, the average longitudinal magnetic field inside the pillar.

(2) Do Landau and Larmor trajectory rotations of electrons about the magnetic field direction occur in the present case?

Reply) The rotation that we obtain might be analogue to that of the of Larmor effect. Indeed, a rotation of a vortex beam is equivalent to a global phase shift. However, our technique relies on the localization of the field to the region near the signal beam. A truly global Larmor rotation would equally rotate the reference beam, and we see no such rotation.

(3) Some experimental measurements involving the interactions between the twisted electron vortex beams and magnetic fields have already been reported (ref 7: Nature, 467, 1839, (2010), ref 26: Nature Commun., 5, 4586 (2014) and ref 31: Nature Physics, 10, 26 (2014).) The differences and advantages in more detail should be included in the

manuscript.

Reply)

- Reference 7 (Nature, 467, (2010)) reports on holographic generation of electron vortex beams and propose a method of studying magnetic dichroism. The method permits one to calculate the out of plane magnetization of a sample, but replication of the results of that measurement have proven challenging [T. Schachinger et al., Ultramicroscopy 179 p. 15, (2017)]. The measurement we report is most similar to electron holography, because it is based on elastic scattering.

- The method reported in Reference 26 (Nature Physics, 10 (2014)) allows the generation of vortex beams, based on interaction with a magnetic monopole which results in an azimuthal phase variation. Indeed it does not consist of a measurement technique. Moreover the value of the OAM produced depends on the in-plane component of the magnetic field.

-Reference 31 (Nature Communications, 5 (2014)) reports on the first observation of Larmor rotation due to interaction of an electron vortex beam with a uniform longitudinal magnetic field (Landau states). This is a beautiful and fundamentally interesting experiment. As discussed in the manuscript, one may think of a strategy to use these rotations to detect the longitudinal magnetic field in real space. A similar strategy is devised towards the end of our manuscript, as we propose a technique based on Larmor rotations with the superposition of OAM carrying electron beams (petal beams). However, the latter may be practically difficult, when the diffraction effects through the material are taken into account.

Upon Reviewer 1's suggestions, we have added the following comments in the main text:

- The method permits one to calculate the out of plane magnetization of a sample
- For Reference 26, we added on page 4: "Recently, it was shown that vortex beams may be generated based on interaction with magnetic monopoles [29]."
- For Reference 31, we added on page 7: "This type of rotation might be interpreted as Larmor rotation that is previously observed for a constant magnetic field [36]."

(4) Relating to the above 1), in page 4, the authors presented the Gouy phase after passing through the monopole, Θ_{Gouy} . The authors should add the theoretical framework and the detailed derivation of Θ_{Gouy} in Methods. Otherwise we cannot evaluate whether the small propagation distance d is a correct assumption and the effectiveness of this method.

Reply) We agree with Reviewer 1 that a detailed derivation of our theoretical model would be useful. We have now included the calculation in the Supplementary Information, i.e. Supplementary Note 1: Derivation of relative phase between positive and negative OAM components, and a second derivation in Supplementary Note 2.

(5) The necessity of the high OAM (200) is unclear. The larger the beam OAM value is, the larger the beam size is. Θ_{Gouy} is inversely proportional to the square of a beam radius ρ m.

Also in page 5, the authors mentioned, “The signal beam interacts with the pillar without being absorbed by the carbon substrate,”. Is the signal beam size larger than the magnetic pillar dimension, 220 nm in diameter? What is ρ_0 in the Fig.4 caption? We are interested what is the smallest possible signal beam size in this method? In addition to responding to these questions in the manuscript, the authors should add the actual beam radius ρ_m and the singular vortex core size.

Reply) The relative phase between the positive and negative OAM component is given explicitly by the formula of $\Delta\Theta$, where one may see that $\Delta\Theta$ depends linearly on the OAM value and thus going to high values of OAM increases the measurable relative phase. Moreover, the beam radius ρ_m here is larger than what can actually be attained as a minimum (not at the diffraction limit here). The size of the electron vortex beam at the focus may be adjusted using the lenses such that its size is comparable to the object under study (pillar). Hence, the beam radius and the OAM value, here, are fully independent. Thus, it is now clear from the formula for $\Delta\Theta$ that increasing the OAM value (independently of ρ_m) is advantageous. We have added the following statement in the main text, to clarify this point:

where ρ_m may be adjusted independently of l .

Regarding Reviewer 1’s second point, the signal beam is tuned to be slightly larger than the transverse dimensions of the magnetic pillar. Hence, the electron vortex passing through the magnetic pillar does experience the phase induced by the pillar. Dimensions for the pillar and for ρ_m can already be found in the main text.

For clarity, we have changed our notation in Fig. 4 from ρ_0 to ρ_m . However, the value of ρ_0 (now ρ_m) for Fig.4 is already mentioned in the caption, i.e. $\rho_0=0.16\ \mu\text{m}$.

Moreover, by defining the singular vortex core as the region where the intensity drops below 5% of its maximum, this area is enclosed in a radius of 10-15% smaller than ρ_m . Nevertheless, this is still much larger than the pillar size. We have added a Figure in the Supplementary Information, Supplementary Figure 2, showing the vortex beam with the magnetic pillar in order to clarify their relative size. Furthermore, the details about the vortex core size have also been added in the caption of the figure.

(6) In page 6, the authors mentioned, “The presence of the magnetic field shifts the position of the bright and dark fringes for both cases.” This is a very key data to give experimental confidence and see the trajectory rotations. The experimental fringe patterns to be distinctly visible should be included in Methods, in addition to the the transverse intensity distributions of the signal beam before-after passing through the magnetic pillar prior to interfering.

Reply) We agree with Reviewer 1 that this data is key here, although it is hard to readily see the shift in the position of the bright and dark fringes. We have thus added this data as Supplementary Figures 3 and 4 and 7 in the Supplementary Information according to Reviewer 1’s recommendation.

(7) An electromagnetic lens employs a strong longitudinal magnetic field. The authors should mention the influence on the measurements and the samples, and the experimental set-up (including Lorentz mode or not, the longitudinal magnetic field of the used lens) in the manuscript.

Reply) The results of our experiment consist in a doubly differential measurement: firstly, with and without the magnetic pillar and, secondly, with a positive and negative OAM values. Hence, the longitudinal magnetic field of the lens will result in a small fixed offset given by the remaining field of the objective, leaving our main results unaltered. Moreover the measurements are linear in the magnetic field. This mechanism permits to remove every contribution except those that depend on OAM. An extended discussion on this point have been added in the Supplementary Information as Supplementary Note 3: External magnetic or electric field. We have also clarified a sentence in the “Experimental Setup” section of the Methods on this point, which now reads,

“We then turned the microscope to the so-called “low mag” mode, with the main objective lens turned to a low strength.”

(8) The magnetic field distribution by the pillar is not uniform. The influence should be mentioned in the manuscript.

Reply) Indeed we show explicitly the experimentally measured magnetic field distribution of the pillar in Figure 2-b. Furthermore, throughout our derivation, it is clear that we are using an extended dipole form, comprised of two monopoles of opposite charges, where at no point the magnetic field is considered constant. However, bound current of the pillar is solenoidal (assuming a constant magnetic flux, one could show that interaction of electron vortex beam with the dipole will give rise to the same phase shift. We have added Supplementary Note 4 describing this last point in more details.

Reviewer 2 - Authors present a new method of measurement of out-of-plane magnetic fields using electron vortex beams in transmission electron microscopy. The experimental results are new and in my opinion convincing. Method can have a strong impact in the field of nano-magnetism and as such I believe that the manuscript is suitable for Nature Communications. I have however some concerns that authors need to address, before the manuscript can be published. They are listed in order of occurrence in the manuscript.

Authors' Response: We thank Reviewer 2 for carefully reading our manuscript and for their insightful suggestions, which helped us improve the quality of our manuscript.

*introductory paragraph: (language) "having l intertwined" -> perhaps find a clearer, more explicit expression?

Reply) We have adjusted the main text accordingly:

“with 1 twisted helical phasefronts”

*page 2, last line: " $\lambda/(2NA)$ " -> the "NA" was defined only as an acronym. How should it be interpreted in a mathematical expression?

Reply) We have modified and adjusted the main text accordingly:

“where in vacuum $NA = \sin \theta$ and θ is the acceptance angle of the lens.”

*page 3, paragraph discussing interaction of a an electron wave with magnetic field omits important references to Edstrom et al., PRL 116, 127203 (2016)Edstrom et al., PRB 94, 174414 (2016)which deal with elastic interaction of electron vortices with magnetic fields in samples - a primary topic of this manuscript. These works must be put into context of the present work, to highlight parallels and differences of the method used by authors and the one proposed in the above-mentioned works.

Reply) We agree with Reviewer 2 that these references are important. They were initially omitted due to a lack of space, but we have now added them to our manuscript according to Reviewer 2's recommendation.

“A theoretical approach using dynamical scattering to measure out-of-plane magnetic fields, without relying on interferometry, was recently proposed [33, 34].”

*page 3, last two lines - some references highlighting the debate of the existence of magnetic monopole would be suitable here

Reply) We have added the following reference to the main text highlighting the debate about the existence of magnetic monopoles.

“Although magnetic monopoles have been simulated in different experiments [25, 26], the search for natural magnetic monopoles has so far been unsuccessful [27].”

*page 4: please define the exact meaning of the "transverse loops" in the definition of “a”

Reply) We have modified the main text defining the meaning of transverse loops.

“the radius of a transverse loop surrounding the object of interest”

*page 4: a reference for the Gouy phase would be suitable at the place, where it is introduced

Reply) We agree with Reviewer 2 that since the Gouy phase is central to our calculation, we should reference it accordingly. We have added an appropriate reference in the main text. Moreover, we have added a Supplementary Note in the Supplementary Information about the derivation of the explicit form of the Gouy phase acquired by the electron vortex beam going through the pillar.

“A particularity of electron carrying OAM is their propagation dependent phase shift known as the Gouy phase shift [30,31].”

*page 4: authors should define the "Rayleigh range" to simplify reading for a reader not familiar with this concept. Alternatively, a suitable reference would be helpful.

Reply) We have added the following definition of the Rayleigh range in the main text and added the appropriate reference.

“ $z_R = \pi w_0^2 / \lambda_{dB}$ is the Rayleigh range and w_0 is a beam parameter given by the beam radius at the waist [30].”

*page 4 (language): "See Method for more details" -> "See Methods for more details”

Reply) We have modified the sentence accordingly.

*page 5: "is a Bessel beam..." - I think it would be fair to write rather "is approximately a Bessel beam...", because there is a non-negligible finite range of radii passing through the aperture

Reply) We agree with Reviewer 2 and have adjusted this sentence according to their suggestion.

*page 5: the 2nd paragraph ("The grating pitch...") I found far too dense. Particularly its second half starting with "The interference pattern...". I would encourage the authors to expand this paragraph and provide a more detailed explanation that would be more accessible to the reader.

Reply) According to Reviewer 2's recommendation, we have modified the section mentioned in order to make it more accessible to the reader.

“The interference pattern at the image plane is a ring of black and white fringes with 200 dislocations. This is due to the fact that the electron wavefunction at the reference arm carries an OAM of 200 with respect to the signal beam. Hence, such a setup is capable of revealing any induced relative phase changes Θ between the two beams. The relative phase changes between the two arms results in a shift of the fringe pattern at the image plane, where a π -phase change shifts the bright fringes to dark and vice versa. The relative phase shift between the beam carrying OAM and the reference beam is obtained by performing the Fourier transform of the interference pattern images in different cases.”

*page 6 (language): "almost height dependent" -> perhaps rather "to some extent height dependent" or something similar?

Reply) We have modified the text according to Reviewer 2's suggestion.

* page 6: "sensitivity of the technique is about $10^8 \mu_B$ " -> authors should explain, how was this estimated

Reply) The present measurement have an error of about 10% of the value , this is therefore the sensitivity that can be further increased by a factor 10 with $L=2000$ that is currently reachable (see also Mafakheri, et al., Applied Physics Letters **110**, 093113 (2017)). The relation between measured magnetic field flux and number of Bohr magneton is the following,

$$B V = \mu_0 m$$

where B is the magnetic field inside the pillar (assumed constant) V is its volume, μ_0 is the vacuum permeability and m is the magnetic dipole moment.

*page 11: can authors elaborate, under what assumptions/conditions does the effect of residual transverse forces vanish in the azimuthal average? Are these conditions fulfilled for the presented experimental setup?

Reply) In order to address this point, we have added Supplementary Note 3: External magnetic or electric Field.

*In addition, I would encourage authors to show the results of measurements for the other pillar quoted in the text - perhaps in Supplementary Information or as a new panel in Fig.4.

Reply) We have added the results of measurements for the second pillar in the Supplementary Information as Supplementary Figure 5: Experimentally observed phase shift of twisted electron beams upon interaction with a magnetic pillar in the second sample.

*Also, a more explicit description of the quantification of the average magnetic field in the sample would be helpful for researchers, which might want to apply this new technique in their experiments.

Reply) We have added some additional detail on the procedure with images of some intermediate steps in Supplementary Note 4 and Supplementary Figure 6.

Reviewers' comments:

Reviewer #1 (Remarks to the Author):

I recommend this paper for publication in Nature Communications.

Reviewer #2 (Remarks to the Author):

Authors have significantly improved the manuscript, addressing most of the questions and comments of both referees. However, the now cited works [36,37] have more in common with the present manuscript than the authors seem to want to admit. In my understanding, both the present manuscript and the mentioned publications discuss using a vortex beam with large orbital angular momentum for detection of out-of-plane magnetization in the elastic scattering regime. Even the approximate proportionality of the effect to the size of OAM is discussed in above-mentioned works. Authors should not only point out the differences (which they did correctly), but also common points of the present manuscript with these works, to put their work in a proper context with published literature.

We would like to thank all reviewers for their thoughtful comments and feedback. Their thoughts helped us to improve the clarity and the quality out work.

Reviewer #1 (Remarks to the Author):

I recommend this paper for publication in Nature Communications.

Reply) Many thanks for your valuable comments and feedback.

Reviewer #2 (Remarks to the Author):

Authors have significantly improved the manuscript, addressing most of the questions and comments of both referees. However, the now cited works [36,37] have more in common with the present manuscript than the authors seem to want to admit. In my understanding, both the present manuscript and the mentioned publications discuss using a vortex beam with large orbital angular momentum for detection of out-of-plane magnetization in the elastic scattering regime. Even the approximate proportionality of the effect to the size of OAM is discussed in above-mentioned works. Authors should not only point out the differences (which they did correctly), but also common points of the present manuscript with these works, to put their work in a proper context with published literature.

Reply) Upon the reviewer's advice, we have added the following passage to the discussion section of the manuscript addressing both the similarities and differences between our work and those reported in refs. [36,37].

“Energy-loss magnetic circular dichroism (EMCD) has been introduced and exploited as a method to reveal magnetic chirality of materials, where the cross section of inelastic interaction of electron beams with a material depends on its magnetization state [36]. EMCD can be evaluated by analyzing the OAM state of the inelastically scattered electrons [20]. However, dictated by the selection rule, only electron vortex beams possessing OAM values of $l = 0$ and $l = \pm 1$ can contribute to the dipole transition, and thus the technique is limited to low-OAM carrying beam within the dipole approximation. Furthermore, methods based on elastic scattering have also been proposed or demonstrated as possible probes of out-of-plane magnetic fields. In particular, the Zeeman interaction of the electron's OAM with the out-of-plane magnetic field in a magnetic material induces a change in the diffraction pattern of the scattered electron [37, 38]. Nonetheless, at the atomic resolution level, using low values of OAM, this technique suffers from detection limitations, but may be feasible with high OAM values. Although in several cases image rotation has been associated with an out-of-plane magnetic field [32, 33, 39, 40], the experiment we present here is the first demonstration of

quantitative measurement of an out-of-plane magnetic field with an electron vortex beam. The interferometric approach we present is a form of off-axis holography and has the advantage of the possibility to quantify and subtract off in-plane magnetic and electro- static effects. Alternatively, image rotation-based methods offer an otherwise similar sensitivity to out-of-plane magnetic fields. The dynamical scattering approach [37, 38] offers a different contrast mechanism and, if experimentally demonstrated, may be a promising alternative. Unlike the technique we demonstrate, this method requires no reference and relies on the combined effect of diffraction and Larmor rotation to provide sensitivity to out-of-plane fields.”

REVIEWERS' COMMENTS:

Reviewer #2 (Remarks to the Author):

I recommend this paper for publication in Nature Communications.